# Spatial Analysis and Socio-Environmental Determinants of Canine Visceral Leishmaniasis in an Urban Area in Northeastern Brazil

**DOI:** 10.3390/tropicalmed10010006

**Published:** 2024-12-26

**Authors:** Natan Diego Alves de Freitas, Lucas José Macedo Freire, Suely Ruth Silva, Nilton Guedes do Nascimento, Pedro Cordeiro-Estrela

**Affiliations:** 1Laboratório de Mamíferos, Programa de Pós-Graduação em Ciências Biológicas, Departamento de Sistemática e Ecologia, Centro de Ciências Exatas e da Natureza, Universidade Federal da Paraíba, João Pessoa 58051-900, PB, Brazil; 2Laboratório Interdisciplinar de Vigilância em Diptera e Hemiptera, Instituto Oswaldo Cruz, Fiocruz, Rio de Janeiro 21040-360, RJ, Brazil; lucasfreire@aluno.fiocruz.br; 3Programa de Pós-Graduação em Medicina Tropical, Fiocruz, Teresina 64001-350, PI, Brazil; 4Gerência de Vigilância e Controle de Zoonoses, João Pessoa 58052-200, PB, Brazil; sulyars@gmail.com (S.R.S.); niltonguedesses@gmail.com (N.G.d.N.)

**Keywords:** kalazar, one health, ecoepidemiology, remote sensing, urban zoonosis, domestic dogs, neglected infectious diseases, vector-borne diseases

## Abstract

The urbanization process has led to significant changes in the landscape, shifting the epidemiological profile of the visceral leishmaniasis (VL) in Brazil. Dogs are considered the main urban reservoir of VL, whose infections precede cases in humans. In order to understand the socio-environmental determinants associated with canine visceral leishmaniasis (CVL), we conducted a spatial analysis of CVL cases in northeastern Brazil from 2013 to 2015, georeferencing 3288 domiciled dogs. We used linear mixed models to understand the ecoepidemiological determinants of CVL spatial relative risk (CVL SRR). Our findings indicate heterogeneity in CVL distribution, with 1 km diameter clusters potentially connected within an estimated 4.9 km diameter by the Ripley-K statistic. In our best-fit model, the CVL SRR was positively correlated with the proportion of households with literate heads, with trees, and with open sewage, but negatively correlated with vegetation phenology and mean income of the census sector. Here, we discuss the potential maintenance source of urban CVL clusters on a One Health framework. These findings highlight the complex interplay of socioeconomic and environmental factors in shaping the spatial distribution of CVL.

## 1. Introduction

Visceral leishmaniasis (VL) is the most concerning form of zoonosis caused by the trypanosomatids of genus *Leishmania* spp. that is transmitted by female phlebotomine sandflies [1]. This vector-borne disease is widely distributed and considered endemic in 80 tropical and subtropical countries [1], though it is more associated with socioeconomic vulnerabilities [2,3], reinforcing the poverty cycle [4].

In the Americas, Brazil was responsible for 96% of more than 59 thousand reported visceral leishmaniasis cases to the WHO between 2013 and 2022 [1]. In this context, *Leishmania infantum* (Ross, 1908) [5] is the etiological agent mainly transmitted by the bite of *Lutzomyia longipalpis* (Lutz and Neiva, 1912) and *Lutzomyia cruzi* (Mangabeira, 1938) [6,7]. Since the 1980s, there has been a notable shift in the trends of VL in Brazil, attributed to the intense rural exodus towards urban peripheries [3,4,8]. This population movement has been unplanned, leading to the establishment of human conglomerates with poor living and household conditions near natural or semi-natural vegetation, thus exposing individuals to the VL sylvatic cycle [4,8,9,10]. In the northeast region, which is characterized by significant wealth disparities, VL does not show signs of significant decrease despite decades of efforts [11,12]; instead, it is stable yet with high risk, evidencing the challenge of controlling the determinants [13].

Changes in the landscape have intensified the close contact between the domestic urban reservoir (the dog *Canis familiaris* Linnaeus, 1758) and the sylvatic (e.g., the crab-eating fox *Cerdocyon thous* (Linnaeus, 1766), the white-eared opossum *Didelphis albiventris* Lund, 1840, and some bat species) [14], the adaptation of the vectors to this new environment may have facilitated the urbanization of the disease [2,15]. However, this expansion underscores the lack of attention, turning visceral leishmaniasis into one of the most significant emerging neglected diseases worldwide [1] and a major problem for human health [16,17].

In order to reduce the transmission rates and the morbidity associated with the VL, the Brazilian ministry of health implemented the program for the Monitoring and Control of Visceral Leishmaniasis in the early 1980s. Along with measures focused on vector control, rapid diagnosis and treatment of humans, and health education interventions, these measures included the culling of seropositive urban canine reservoirs [18]. Despite the efforts with the strategies applied, the disease expanded into new areas [19,20] and the number of cases increased [21]. In particular, the practice of canine culling has been identified as a controversial and inefficient control measure [22,23]. Since studies have observed a spatial correlation between canine and human cases, where cases in dogs precede infections in humans [18,24], new approaches are needed to understand the epidemiological factors associated with infections in dogs.

Spatial analysis has become a cornerstone in the surveillance and control of leishmaniasis in Brazil [25,26], enabling the identification of priority areas and guiding interventions. The Brazilian Ministry of Health has a surveillance program aimed at identifying and monitoring areas of epidemiological significance for control measures, like environmental and socioeconomic features [27]. This field of science focuses on understanding the geographical clustering patterns, as well as the factors influencing disease occurrence [28]. In Brazil, spatial data analyses have been extensively used at different scales to study VL, providing valuable insights into disease spatial patterns [4,25,26] and risk factors [2,3,29]. Understanding these spatial dynamics is essential for the effective implementation of prevention and control programs, ensuring resources are optimized to tackle the most at-risk areas.

The Brazilian northeastern region concentrates the majority cases of VL in Brazil [11], and the Paraíba state is one of the areas which lacks studies about the ecoepidemiology of this disease in humans and dogs. The municipality of João Pessoa, capital of Paraíba, represents a different model for the study of VL since its successional stage in the urbanization processes is significantly delayed in comparison with the majority of Brazilian state capitals that went through this stage in the late 20th century. Furthermore, it has more than half of VL cases in Paraíba [30], and the VL vector, *L. longipalpis*, is distributed along the urban matrix of this city [31]. Despite that, little is known about the spatial determinants of the VL in this city, especially in dogs; cases in these animals precede cases in humans [18]. Therefore, the present study aims to analyze the spatial patterns and socioeconomic and environmental determinants of canine visceral leishmaniasis between 2013 and 2015 in João Pessoa, the capital of Paraíba state, in northeastern Brazil.

## 2. Materials and Methods

### 2.1. Study Area

The municipality of João Pessoa (07°07′12″ S 34°52′48″ W) is the capital of the state of Paraíba, with an area of 211.475 km^2^, an estimated population of 833,932 inhabitants, and a demographic density of 3424 inhabitants/km^2^, comprising 73 neighborhoods [32]. João Pessoa has a tropical climate with dry summer, according to the Köppen classification for Brazil [33], with its rainfall period between March and August, 1500 to 1700 mm/year, as well as an annual average of 25 °C and relative humidity of around 80% [34]. As part of the coastal region of Paraíba, João Pessoa is situated within the Atlantic Forest domain [34], which is a biodiversity hotspot [35], with a 44% of the municipality (93.2/211.5 km^2^) still covered by the semideciduous seasonal forest type, mangroves, riparian forests, and the tabuleiros formation, which is characterized by grassland trees [34]. The urban matrix development occurred heterogeneously over an intensive reduction and fragmentation of the Atlantic Forest from 1970 to 2010 [36,37,38]. In this regard, João Pessoa is a coastal, medium-size city very representative of the world scenario of urbanization in developing countries [39].

### 2.2. Surveillance Data and Geoprocessing

The data utilized in this study were provided by the Environmental Surveillance and Zoonosis Management Center (ESZM) of João Pessoa. These data were obtained from the canine visceral leishmaniasis survey in domiciled dogs between 2013 and 2015. The survey followed the operational protocol of the Zoonoses Surveillance and Control Manual [18]. In 2013, the Municipal Secretariat of Health randomly delimited random blocks within districts, following the protocol of the rapid survey of the *Aedes aegypti* infestation index—LIRAa [40]. After identification of a positive case of CVL, the ESMZ conducted visits to all houses on the street, revisiting the blocks in the subsequent two years. The dogs were initially tested for infection using the Rapid Test for Diagnosis of Canine Visceral Leishmaniasis (TR DPPH^®^, Bio-Manguinhos, FIOCRUZ, Rio de Janeiro, RJ, Brazil). The dogs that tested seroreactive were confirmed with the Enzyme-Linked Immunosorbent Assay (ELISA, Bio-Manguinhos, FIOCRUZ, Rio de Janeiro, RJ, Brazil) in accordance with the recommendation of the Brazilian Ministry of Health. The CVL prevalence was calculated annually by dividing the number of seropositive dogs by the total number of dogs screened.

The dogs from the CVL survey were retrospectively georeferenced by searching for their address on the surveillance forms in a geographic information system (GIS) tool. In cases in which the address was not written precisely or not found, we shared a GPS with ESZM agents to localize the dogs’ coordinates in situ. This study included only dogs that were domiciled and for which the owner had consented to the sampling by the surveillance service. The georeferencing of the data (sampling of notification forms) was random. The number of forms was not the same for positives and negatives (proportion of sampled forms from the total and total number of forms sampled) because there were many more negative results than positive ones. The purpose of sampling the negative forms was to test whether areas without reactive dogs were non-reactive due to a lack of sampling or a true absence. Consequently, approximately 20% of the total sample was included, which involved a considerable effort of georeferencing approximately 2000 addresses.

### 2.3. Spatial Analysis

To estimate if there are clusters of domiciled CVL and the radius of clusters, we used the Ripley-K function (Kest or reduced second moment function) as implemented in the spatstat package in R [41]. The function infers the presence of spatial structure and estimates aspects of dependence among cases by comparing the intensity of events to a random (Poisson) point process. The definition of the Kest statistic is:$$Kest\left(r\right)=\left(\frac{a}{n.\left(n-1\right)}\right)*\sum \left[i,j\right]I\left(d\left[i,j\right]⩽r\right)\left(e\left[i,j\right]\right)$$
where *r* is the influence range of the pattern, *a* is the area delimited by the latitude and longitude of the municipality, and *n* is the number of CVL occurrences. The sum is taken from all the ordered pairs of points *i* and *j* in the observed point pattern. The distance between two points is given by *d*[*i*,*j*], and *I*(*d*[*i*,*j*] *≤ r*) is the indicator if the distance is less than or equal to the influence range. The term *e*[*i*,*j*] is a correction due to occurrences outside of the study area not included. We implemented the border correction to reduce edge effects from the unobservability of these points [41].

We explored case clustering by estimating the spatial relative risk of canine visceral leishmaniasis (SRR CVL) cases. We used the kernel smoothed intensity (density.ppp) function from spatstat package in R [41]. The function calculates the intensity of spatial events, assigning decreasing weight with increasing distance [42], within the estimated bandwidth by the Scott method [43] with the bw.Scott function of the same package. This estimation considers the cluster as anisotropic, resulting in a separate bandwidth to each coordinate axis. We extracted the CVL density and seronegative dogs’ density around each sampled dog and calculated the spatial relative risk by dividing the CVL density by the density of the seronegative dogs. The SRR CVL was used as a continuous response variable in our models.

### 2.4. Environmental and Socioeconomic Determinants

Vegetation phenology was quantified via remote sensing, using the Enhanced vegetation index (EVI) as a metric for calculating the greenness of vegetation. This was achieved by combining the spectral bands captured by satellite sensors, as illustrated by the following equation as
$$\mathrm{EVI}=2.5*\left(\mathrm{SBnir}-\mathrm{SBred}\right)/\left(L+\mathrm{SBnir}+C1*\mathrm{SBred}-C2*\mathrm{SBblue}\right),$$
where *SB* refers to spectral bands for near infra-red (*nir*), *red*, and *blue*; *L* is the soil adjustment factor; while *C1* and *C2* refer to the aerosol resistance. The values of *L*, *C*1, and *C*2 in the EVI formula for the MODIS and Landsat data are 1, 6, and 7.5, respectively. This index was identified as an environmental determinant due to its potential to serve as a proxy for the presence of vectors, the sylvatic reservoirs (crab-eating fox, crab-eating raccoon, small mammals, and bats), and adequate oviposition sites [18]. We used EVI instead of NDVI, since it gives a more accurate picture of the heterogeneity of vegetation [44], which might be especially important in cities with parks, orchards, natural, reforested areas, and, more generally, cities with seasonal climates. EVI allows spatio-temporal variations from the biophysical and structural properties of vegetation to discriminate through the spectral response of chlorophyll [45]. EVI also appropriately corrects cloud effects and spectral chlorophyll values in tropical forests [46]. We selected scenes from Landsat 8, with a spatial resolution of 30 m and temporal resolution of 32 days from the 2013–2015 period. We obtained a series of satellite images courtesy of the U.S. Geological Survey (USGS) through the Google Earth Engine platform. We calculated the vegetation phenology in peridomicile by the mean of EVI values between 2013 and 2015 and the census sector vegetation phenology with the mean values inside each census sector polygon.

We considered socioeconomic determinant data obtained from the 2010 census of the Brazilian Institute of Geography and Statistics (IBGE). The IBGE census database is frequently used as the layer of human population data [4,15,26,47,48,49,50,51]. We calculated the proportion of households with a specific variable of interest (e.g., households with open sewage) by dividing it by the total number of households in the census sector to adjust for differences in the census sector size. We removed the auto-correlated variables as the proportion of households with electric energy, public illumination, and garbage collection and inappropriate garbage disposal, through the Variance Inflation Factor (VIF) by the Kendall method and stepwise procedure, using the usdm package [52]. After remotion, we maintained seven variables: proportion of households with accumulated garbage, proportion of households with income less than one minimum wage, proportion of households with literate heads, log of the mean income, proportion of households with open sewage, proportion of households with paved streets, average residents per household, proportion of households with trees, and proportion of households with inappropriate environmental sanitation in the census sector. These variables and their biological interpretation for CVL are detailed in Table 1.

### 2.5. Statistical Analysis and Model Selection

We used linear mixed-effects models (LME) to analyze the degree to which socioeconomic and environmental variables were associated with the estimated SRR CVL. These variables were treated as fixed effects, while the code of each census sector was considered a random effect, as most of the variables were nested at this scale. We compared the models with single and multiple variables and selected the best-fit models by the Bayesian Information Criterion (BIC). The analyses were conducted in an R environment (version 4.4.0) using the nlme package [53], and the model selection was automatized with the dredge function of the MuMIn package [54].

## 3. Results

### 3.1. Survey Data and Mapping

The ESZM notified us of 1458 dogs infected with *L. infantum* out of the 11,742 sampled during the 2013–2015 surveillance period (Table 2).

We georeferenced 3288 domiciled dogs, corresponding to 68.4% and 19.5% of the total serum reactive and negative serological results, respectively (Figure 1).

On average, 50.9 ± 74.5 domiciled dogs were tested per district, indicating a skewed sampling representation. The dogs’ tested prevalence varied widely across districts, ranging from 0% to 87.62%, with an average of 15.4 ± 24.11 cases per district (Figure 2). Eight districts remained free of occurrences throughout the study period.

### 3.2. Spatial Analysis

Our spatially explicit analysis revealed three main clusters in the estimated densities of canine visceral leishmaniasis (dCVL) (Figure 2). Ripley’s K function indicated a significant deviation from homogeneity (*p* = 0.002), with CVL cases clustering within an estimated 4.9 km diameter. The estimated clustering of dCVL by the Scott method had a diameter of 1.1 ± 0.1 km, ranging from 0 to 14.9 per km^2^, averaging 3.2 ± 5.9 dCVL per km^2^ from the ESZM test database.

The comparison of the categorical analysis of prevalence against the relative risk analysis of point patterns highlight very different perspectives. The high prevalence of the district above the west cluster (Figure 2A) was driven by low sampling as shown in Figure 2B. Districts with high relative risk clusters showed a mid-range prevalence, despite being at the cores of the three main clusters. The coastal districts in the southeast exhibited high prevalence due to their proximity to the two main clusters and concentrated cases surrounded by degraded native vegetation. The sampling outcome is presented in the map in Appendix A, which depicts the density of reactive and non-reactive dogs. It can be observed that the density of negative dogs is largely consistent outside of the foci, which suggests a random sampling. Additionally, it can be observed that the higher density of the negative dogs is similar to the density of the positive dogs, which, according to the LIRAa protocol, are the foci, i.e., the most densely sampled areas.

### 3.3. Statistical Analysis and Model Selection

The best-fit explanatory model of CVL spatial relative risk included the following variables: mean values of vegetation phenology in peridomicile, proportion of households with literate heads, proportion of households with trees, proportion of households with open sewage, and log of mean income in census sector. This linear mixed model incorporating these five significant variables (logLike: 915.9, BIC: −1765.8) differed by 4.2 BIC units from the second-best model, which includes the proportion of households with paved streets in the formula composition (Table 3). The detailed comparison of models and variables’ estimated effects are included in Appendix A.

Variables included in the best-fit as the proportion of households with trees (r = 0.7 ± 0.1, df = 611, t-value = 4.8, *p* < 0.01), the proportion of households with open sewage (r = 0.6 ± 0.1, df = 611, t-value = 5.1, *p* < 0.01), and the proportion of household with literate heads (r = 3.3 ± 0.5, df = 611, t-value = 6.7, *p* < 0.01) exhibited a positive correlation with the CVL spatial relative risk, while the vegetation phenology in peridomicile (r = −0.3 ± 0.04, df = 2670, t-value = −8.4, *p* < 0.01) and the log of mean income of the census sector (r = −0.8 ± 0.1, df = 611, t-value = −13.5, *p* < 0.01) were negatively correlated (Figure 3). The random effects related to each census sector was responsible for 52% of the variance unexplained by the fixed effects in the model.

## 4. Discussion

The spatial relative risk of canine visceral leishmaniasis (CVL) cases are heterogeneously distributed in the city of João Pessoa, and predominantly centered in the south. The county of João Pessoa experienced an unregulated and swift urbanization process towards the periphery of the city, particularly in the southern district. This was precipitated by the rural exodus of populations and the deforestation process, ultimately leading to the establishment of urban conglomerations marked by substandard residential conditions [34]. Geoprocessing canine samples provided unique insights compared to the district-based prevalence approach used for targeted control or preventive interventions. These clusters emerged in districts marked by socioeconomic and environmental factors closely linked to substandard urbanization and modifications of native vegetation. Typically found on the outskirts due to its presence around Atlantic Forest fragments, this vegetation appears as degraded native vegetation or backyard orchards.

### 4.1. The District Prevalence and Continuous Spatial Analysis

The census sector division is the usual approach for evaluating the epidemiology of leishmaniasis [4,15,24,26,48,49,50,51,55,56,57,58,59]. This approach is rooted in the health system’s territorialization shaped by political, socioeconomic, and cultural interests, along with district similarities [60]. However, a continuous spatial analysis can delineate target areas independently of administrative boundaries [61], thus providing a more precise understanding of the urban ecosystem’s heterogeneity [28,62,63] compared to summarized aggregated data [25]. Additionally, the sample size biases the prevalence approach; hence, districts with a small number of tested dogs and a higher seropositive count might falsely indicate a high prevalence in a non-priority zone due to the inherent unbalanced sampling design of surveillance schemes [28,50]. Given the heterogeneity in occurrence and distribution across districts, employing a spatial analysis helped delineate clusters for intervention [61]. Therefore, methods that extrapolate beyond limits provide a shift from a sectional perspective to an ecological one [15,56,63].

### 4.2. Is Cluster Size Informative of Maintenance Mechanisms?

The estimated distance of cases clustered (Kest) of 4.9 km suggests that the hotspots of CVL density are connected, each with approximately 1.1 km of range in three districts. This emphasizes that the cluster size by itself might be informative of the maintenance mechanism. Estimations of case density typically rely on the small flying range (~300 m) of phlebotomine vectors [47,51,56,64,65,66,67]. However, when the radius of influence is not predetermined, kernel density usually estimates a larger clustered distance [2,68], similar to our Kest results. Here, this distance is too large to suggest the alternative hypothesis that the scale of the cluster is determined by urban terrestrial small mammals (the white-eared opossum for example), which typically have ranges lying around 100–500 m [69,70]. Another hypothesis suggests that bats, which are abundant in neotropical urban landscapes and roost in houses and buildings [71], are a determinant of the density of CVL. However, the scale of the cluster is too small for most bats [72]. In our perception, the estimated range of clusters suggests that stray dogs might be maintaining spatial clusters of CVL in domiciled dogs. The estimated distance by Kest aligns more closely with the home range of stray dogs [73,74,75] than vector dispersion [76,77,78] in an urban environment. Stray dogs may be acting as bridge hosts [79] in urban ecotone environments. Stray dogs often live in close proximity to humans and domestic and wild animals, and this relationship can negatively impact human health due to an increased risk of transmission and maintenance of zoonoses, including visceral leishmaniasis (VL) [80,81,82]. The prevalence of this disease is higher in stray dogs compared to domiciled ones [83], especially in areas with outbreaks of this disease [84]. These animals tend to have a large living area [74], and they can come in contact with a variety of locations conducive to the presence of sand flies and become infected with the etiological agent of VL. Moreover, stray dogs can introduce this etiologic agent into new areas where the vector lives [85], especially in urban settings where the vector is present in high densities [31], once dogs are considered the main urban reservoir of VL [86]. Nonetheless, the determinants of the spatial disease-ecology of free-ranging dogs and wildlife are mostly unclear due to financial limitations. To better understand this hypothesis, the surveillance of stray and domiciled domestic animals is indispensable [47,63], as well as that of wildlife species and humans, in high-risk zones, particularly in urban ecotones.

### 4.3. Socio-Environmental Determinants of Canine Visceral Leishmaniasis

Our best-fit linear mixed effects model by BIC shows that the spatial risk of CLV is strongly associated with unplanned and recent landscape occupation. The determinants of higher risk suggest inefficient habitation management, associated with an unhealthy condition to dogs, humans, wildlife, and the environment, a clear One Health-type problem. Our findings indicate that the proportion of households with trees, with open sewage and, with literate heads were positively associated, while mean income and vegetation phenology of peridomicile were negatively associated with a higher risk of CVL. This proposes that the disease is related to peripheral areas of the county exposed to environmental changes. Landscapes undergoing urbanization over poor socio-environmental conditions can disrupt the ecology balance [87] and increase the risk of VL’s infections [88]. Cities that went through an intensified and unplanned urbanization process towards the periphery create environments suitable for the establishment of the VL cycle, resulting in dogs living in these areas having twice the risk of VL infection compared to those in better socioeconomic conditions [89,90]. It is important to highlight that a heterogeneous urban process produces idiosyncratic situations such as districts with luxurious condominiums without paved streets, domiciles with public illumination but without appropriated sanitation, places with illegal occupation with semi-natural arborization and dogs widely present.

Other Brazilian capitals observed this zoonosis having higher incidence in areas with poor living conditions and recent urban expansion [3,4,15,51,88,91,92,93]. Regarding the association between the literate proportion in the census sector and the disease, our findings may appear to be in disagreement compared to the existing literature [3,29,94]. We hypothesize that this discrepancy could be attributed to literate individuals having greater access to healthcare information [64]. This group may be more likely to recognize the signs of illness or be more attentive to their animal’s health, leading to a higher likelihood of seeking veterinary care and diagnostic exams [95,96]. However, the lower mean income is an indication that the group probably knows less about the VL cycle [95] and is less supportive of the use of preventive methods due to cost [97,98]. This combination is concerning, as animal illness is a major factor in the abandonment by families that do not have the income to treat them [99]. The abandonment of an ill animal and its substitution by a newer one, commonly immunologically naive [100,101,102], is a source of maintaining clusters.

The structural effect of vegetation on risk remains controversial, showing positive, or negative, or even neutral patterns in some cases [29]. Vegetation serves as a proxy for indirect effects that are costly to quantify, such as surveys in wildlife hosts and vectors, and the habitat and resources of cycle participants. In certain cases, areas with positive association between vegetation indices and CVL present a rural–urban transitional pattern of transmission, with suitable conditions for the sandfly vector and proximity to forest fragments and the sylvatic cycle [67,103]. In contrast, areas with negative association represent the urban matrix, where the transmission pattern occurs in the urban matrix where the vector is fully adapted in those areas [9]. This variable was measured in different methods and scales in the context of VL [26,29], including remote sensing techniques. Studies that explored this influence observed more cases in sparse vegetation than dense ones [104,105,106] or multivariate models detect a significantly higher risk of vegetation only in interacting areas with growing population rates [15]. These observations suggest that an environment’s characteristics and the type of vegetation are more important than its mere presence or absence.

We attribute the risk modulation by vegetation in two different ways: positively as trees in households and negatively with the vegetation phenology in peridomicile. Lower values of EVI indicate a simplified vegetational stratum, predominantly herbaceous types, shrubs, and grass, while the opposite means a higher forest canopy and healthy hydric condition [45]. The inverse correlation between the proportion of households with trees in census sectors and the vegetation phenology in peridomicile may indicate changes in vegetation compositions within households. Census sectors with a higher proportion of households with trees tended to have lower vegetation phenology values. Also, the definition used by IBGE for trees in households is ambiguous, potentially including remnants of the Atlantic Forest, exotic tree species, and backyard orchards. In this context, the proximity of the VL sylvatic cycle by spillover events is intensified [107,108] due changes in the landscape’s ecological interactions [79,87]. It is necessary to understand how the development of the socio-vulnerabilities in degraded environments influences the sylvatic cycle in order to implement effective political interventions within a One Health approach.

Ensuring coexistence among human populations, domestic animals, and wildlife, biodiversity is crucial for sustainable development. In the context of leishmaniasis in peri-urban areas, promoting One Health approaches that integrate human, animal, and environmental health can enhance surveillance and control efforts. The Health Ministry’s guidelines emphasize the significance of understanding the ecoepidemiological patterns in Brazilian municipalities vulnerable to VL [27]. These findings underscore the contribution of dog population surveillance in identifying priority areas to concentrate efforts. The CVL clusters highlighted key zones for developing strategies as the increasing the local income, including health educational campaigns, entomological monitoring, the identification of suitable environments for vector reproduction, and the coverage of the dog population with insecticide-impregnated collars [27] and implementing sustainable land-use practices and reforestation efforts to restore degraded habitats [36]. Furthermore, we highlighted the importance of the dog owner in the prevention of CVL, where there is a reduction in the rate of canine infection associated with better care behaviors [109]. These findings show the importance of a spatial analysis, evidencing that interventions need to be coordinated even across distinct districts and independently of administrative barriers [25,110]. The control of VL requires a holistic approach due to its complex nature, involving the interplay between humans, vectors, domestic and wild animals, and the environment [22,23]. Therefore, these interventions need to involve collaborative efforts between health professionals, veterinarians, and environmental scientists to address disease transmission at the human–animal–environment interface.

There are several limitations to consider in this study regarding the interpretation of the surveillance data and socioeconomic variables. Firstly, our findings assume that the infections occurred in the household environment, which may not always be the case [25,61]. Secondly, the geoprocessing using secondary data collected by health services lacks spatial sampling delineation for this purpose. Our study used secondary data from municipal zoonosis surveillance, following the LIRAa protocol recommended by the Brazilian Ministry of Health. The sampling of dogs was performed through random block assignment, with all houses on the street being sampled if positive dogs were found. The protocol detects and delimits foci, although it is not completely random. Within the foci, the sampling is more thorough, until completely negative streets are found. This leads to the following consequence: once the relative risk (positives/negative) is considered and these higher risk zones are located within the foci—which are thoroughly sampled using the same protocol—our risk estimate is likely unbiased. Furthermore, only those households that consented to the sampling process conducted by the ESMZ were included in this study. Refusals were not provided to us, but it is possible that this data may alter the proportion of relative risk without altering the main results and conclusion. Although we showed a significantly higher relative risk in peripheral zones, these areas were mostly undersampled due to limited financial and human resources. The inability to identify addresses a posteriori or in situ, such as farms or planned streets, contributed to the unfeasibility of geoprocessing in these areas. This population likely has less access to veterinary health services compared to households with literate heads, which could partially explain the controversial result. Moreover, socioeconomic characteristics that occurred after the 2010 census are unavailable for comparison using this approach, as observed in other studies [15,47]. Additionally, our interpretations primarily focused on the surveillance of domiciled dogs and did not include stray dogs, which may underestimate the prevalence and risk in the dog population [47,63].

## Figures and Tables

**Figure 1 tropicalmed-10-00006-f001:**
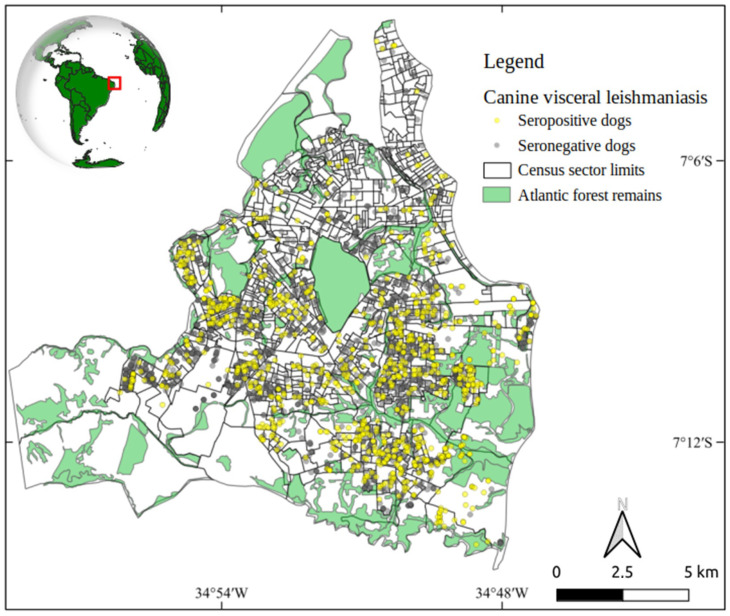
Spatial distribution of serological results of canine visceral leishmaniasis in domiciled dogs in João Pessoa, Paraíba state, Brazil (2013–2015).

**Figure 2 tropicalmed-10-00006-f002:**
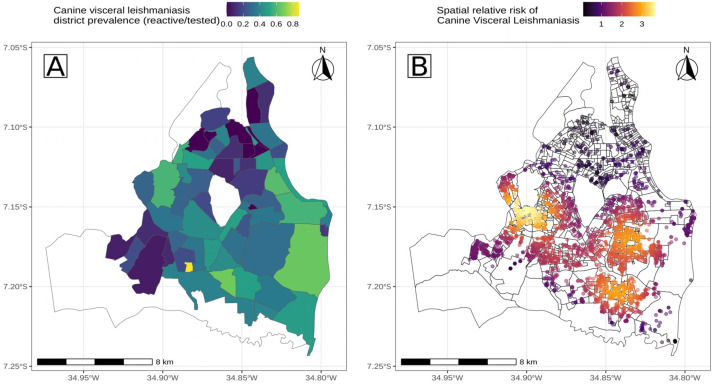
Prevalence (**A**) and spatial relative risk (**B**) of domiciled canine visceral leishmaniasis from the surveillance of households from 2013 to 2015.

**Figure 3 tropicalmed-10-00006-f003:**
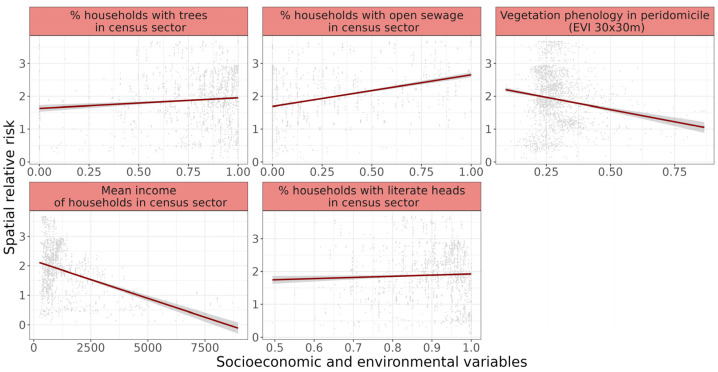
Relationship between the canine visceral leishmaniasis spatial relative risk and socioeconomic and environmental variables. In data, each dot is a domiciled dog, the red line represents the significant trend fitted by a linear mixed model, with the gray zone indicating the 95% confidence interval values.

**Table 1 tropicalmed-10-00006-t001:** Metadata of socio-environmental variables generated from IBGE 2010 census and remote sensing.

Variable (Abbreviation)	Biological Meaning
Average residents per household	Households with higher number of residents may attract the vector due to greater disponibility of blood sources [29].
Proportion of households with income less than one wage	Higher proportion of households with income less than one wage are associated with areas with poor socio-environmental and household conditions, creating suitable areas for vector establishment [29].
Mean income in the census sector	Census sectors with lower mean income may reflect areas with poor socioeconomic conditions, inadequate household structures and conglomerate of human dwellings, associated with suitable areas for vector presence in high densities [29].
Proportion of households with inappropriate environmental sanitation	Associated with greater availability of areas with suitable environmental conditions, like reproductive sites and food sources, for the vector [29].
Proportion of households with literate heads	Illiterate homemakers receive less instruction on accessing health orientation and know less about disease risk [29].
Proportion of households with paved streets	Associated with recent occupation limits; may indicate an unplanned expansion, where wildlife contact is more accessible and probably affected by deforestation; relationship with the vector’s biology, since more paved streets lead to higher local temperatures which negatively impact the vector’s reproduction rate [29].
Proportion of households with trees	Presence of trees may be a source of organic matter and shelter to vectors [29].
Proportion of households with open sewage	May suggest insufficiently protected human rights; associated with unhealthy conditions for dogs (exposure to other pathogens); may provide reproductive sites and food sources for the vectors [29].
Proportion of households with accumulated garbage	May suggest an inefficient governmental management; associated with unhealthy conditions for dogs (exposure to other pathogens); may provide reproductive sites and food sources for vectors [29].
Vegetation phenology	Lower values of Enhanced Vegetation Index (EVI) represent areas with low vegetation cover, associated with degraded areas due to urbanization where the vector (*Lutzomyia longipalpis*) is highly adapted [10].

**Table 2 tropicalmed-10-00006-t002:** Canine visceral leishmaniasis surveillance in João Pessoa-PB, Brazil, 2013–2015.

Year	2013	2014	2015	Total
Dogs sampled	5908	2954	2880	11,742
Seroreactive dogs	263	521	674	1458
Prevalence (seroreactive dogs/total sampled)	4.45%	17.64%	23.4%	12.4%
N of seropositive dogs georeferenced (%)	172 (65.4)	412 (79)	414 (61.4)	998 (68.4)
N of seronegative dogs georeferenced (%)	511 (9)	956 (32.4)	823 (28.6)	2290 (19.5)

**Table 3 tropicalmed-10-00006-t003:** Top five best-fit linear mixed models of determinants of the canine visceral leishmaniasis spatial relative risk in João Pessoa-PB, Brazil, during the 2013–2105 surveillance campaigns as predicted by socioeconomic and environmental variables.

Model	Formula	df	log-lik.	BIC	ΔBIC
1	CVLSpatialRelativeRisk~EVIinPeridomicile + %HouseholdswithLiterateHeads + log (MeanIncome) + %HouseholdswithOpenSewage + %HouseholdswithTrees + (1 | CodeCensusSector)	8	915.3	−1765.8	0.0
2	CVLSpatialRelativeRisk~EVIinPeridomicile + %HouseholdswithLiterateHeads + log (MeanIncome) + %HouseholdswithOpenSewage + %HouseholdswithTrees + %HouseholdswithPavedStreets + (1 | CodeCensusSector)	9	917.2	−1761.6	4.2
3	CVLSpatialRelativeRisk~EVIinPeridomicile + %HouseholdswithLiterateHeads + log (MeanIncome) + %HouseholdswithOpenSewage + %HouseholdswithTrees + EVIinCensusSector + (1 | CodeCensusSector)	9	916.0	−1759.1	6.7
4	CVLSpatialRelativeRisk~EVIinPeridomicile + %HouseholdswithLiterateHeads + log (MeanIncome) + %HouseholdswithOpenSewage + %HouseholdswithTrees + EVIinCensusSector + %HouseholdswithLess1MinimumWage + (1 | CodeCensusSector)	9	915.6	−1758.3	7.5
5	CVLSpatialRelativeRisk~EVIinPeridomicile + %HouseholdswithLiterateHeads + log (MeanIncome) + %HouseholdswithOpenSewage + %HouseholdswithTrees + EVIinCensusSector + %HouseholdswithAccumulatedGarbage + (1 | CodeCensusSector)	9	915.3	−1757.8	8.0

df, degrees of freedom; log-lik., log-likelihood; BIC, Bayesian information criterion.

## Data Availability

The data presented in this study are available on request from the corresponding author.

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
