# Peer review of "Spatial Analysis and Socio-Environmental Determinants of Canine Visceral Leishmaniasis in an Urban Area in Northeastern Brazil"

_tropicalmed, 2024, doi:10.3390/tropicalmed10010006_

Round 1
Reviewer 1 Report
Comments and Suggestions for Authors
The article addresses a relevant topic. However, it has limitations that need to be considered and analyses that need to be added to assess the validity of the presented information.
Main comments
-There is a clear difference in the proportion of reactive and non-reactive dogs that were georeferenced, which may introduce bias that would invalidate the analyses. On the other hand, if the distribution of losses (in this case, the proportion of non-georeferenced animals) was random in space, there would be no bias, or the influence would be minor. Therefore, the authors should explore this deeply, with maps and/or flowcharts and/or tables. These should demonstrate that the spatial distribution of losses, starting from the initial number of animals examined in each block/district of the territory, did not occur differentially. In other words, was the distribution of the proportion of georeferenced dogs and non-georeferenced dogs, reactive and non-reactive, similar between blocks and districts? If this is not the case and the losses were not random, the presented results will not be valid.
-It is also important to highlight the existence of refusals. Authors must describe how many occurred and how they were divided by blocks/districts. Although the results provide this information in summary, a better breakdown is important, mainly because this is also a key factor for the validity of the conducted statistical methods.
-Considering the adopted analytical procedures, the quantity of observations is also a very relevant factor. Despite the large number of animals examined, the number of georeferenced ones is low. The authors should analyze (through the results of the models) and discuss the implications of this.
-It should be specified in the text what was defined as "alarming" (page 3), and it is also necessary to better explain how the division by blocks and districts is done and how they are constructed. The blocks and districts studied and the quantity of animals in each should also be presented spatially.
-Since it is a study with spatial analyses, the inclusion of maps is essential for understanding the methods and for proper visualization of the results. Thus, these should be included. I emphasize that in the version I reviewed, there are no figures, although these have been mentioned in the text.
-Overall, a comprehensive review of English writing and text structuring is necessary. The logic of paragraphs and the order of ideas from different sections, especially in the introduction and discussion, need correction. The writing, in its current format, often compromises understanding.
-The title of the article implies an analysis of the Northeast, but in reality, only the municipality of João Pessoa is analyzed. Therefore, it should be modified.
-The procedure "We normalized each variable by the proportion of households in the census sector to account for census sector size variation" needs to be better explained. How was this calculation done?
Minor comments
-The abstract should present the results of the associations more clearly and didactically.
-It is mentioned in the introduction that visceral leishmaniasis has no downward trend in the Northeast. However, this contrasts with the situation that has been identified in recent years. I suggest consulting and citing data from the Ministry of Health and the article: DOI: 10.1111/zph.13092.
-The introduction does not highlight the study's differentials compared to what has already been published in terms of spatial analyses of canine visceral leishmaniasis.
-I suggest informing the population size of João Pessoa, along with other sociodemographic information.
-Table 2 presents an excessive amount of acronyms, which hampers understanding. It is possible that the acronyms are replaced by full terms and that the footnote is used only for additional explanations.
Comments on the Quality of English LanguageProvided in the "Comments and Suggestions for Authors
" section.
Author Response
Thank you very much for taking the time to review this manuscript. Please find the detailed responses below, along with the corresponding revisions and corrections. Please find the detailed responses below in the attachment, along with and the corresponding revisions and corrections.

Reviewer 2 Report
Comments and Suggestions for Authors
Review of “Spatial analysis and socio-environmental determinants of urban canine visceral leishmaniasis in Northeastern Brazil”
The authors provide a spatial analysis of leishmaniasis cases in dogs in Northeastern Brazil and associated environmental risk factors for the disease.
The manuscript I was provided did not include any figures despite the text referencing those, so I can’t comment on their merit.
Abstract
Line 23, what are “literacy heads”? Presumably this refers to literate heads of household and/or homemakers, please clarify.
Line 24, “vegetation condition” seems very vague, please define what specific condition was the risk factor.
As a general comment, please include some of the conclusions regarding epidemiology/epizootiology with wild mammals that you are mentioning in the discussion (line 289 and following)
Introduction
Line 40, I’d use “accounted for” instead of “was responsible”.
Line 51, I would not necessarily assume the reader to be familiar with what species constitute the sylvatic canine reservoir here, please list them.
Line 63/4, this sentence appears to be incomplete.
Overall, I am missing a rationale as to why the authors chose this particular area for their study, please make this clearer in the introduction.
M&M
Line 102, what does “totally sampled” mean?
Line 112, make it clearer from the beginning that you were only using “domiciled dogs”. Did you look into whether these were kept indoors or outdoors? Why did you not include stray/feral dogs, which presumably are less likely to receive treatment for CVL infection? What did you do to assure that you didn’t sample the same dog multiple times?
Line 141, I am assuming many of the readers will not be familiar with the EVI, please provide more background on what this measures.
Line 160, please list which indicators you removed and based on what statistics you did so.
Line 162 and following, what does “less than one wage” and “homemaker” mean specifically? Is “trees in household” measuring the presence of trees as a yes/no variable, or the number of trees on the property? It would probably be a good idea to transform table S1 into a table included in the main manuscript to better address these questions.
Line 177, indicate your R version and software used.
Results
Line 180, provide the corresponding percentage of infected dogs.
Line 185, why didn’t you georeferenced all dogs?
Line 193, provide the statistics to show whether this difference was significant.
Line 213, “head of family” seems in conflict with “homemaker” above, please ensure consistency. The two do not mean the same thing in English.
Line 211 and following, this implies that you were testing multiple models. Please expand on what you started with in your initial model(s) and how exactly you arrived at the final model.
Table S1 again doesn’t distinguish between “homemaker” and “head of household”, which are two different things in English.
Discussion
Line 250, this seems in conflict with your previous statement in the M&M section that the city of João Pessoa experienced a slow and gradual urbanization process.
Line 291 and following, again, you need to discuss why you only used domiciled dogs and what limitations this presents for your conclusions. Did you collect any data on the number of stray dogs in these spatial units? If not, why not?
Line 331, is this for Brazilian State capitals or what exactly do you mean by “Brazilian capitals”?
Line 342, include a statement regarding the animal welfare implications of animal abandonment.
Line 345 and following, explain what specific mechanisms you think lead to changes in vegetation being associated with higher or lower CVL risk.
Comments on the Quality of English Language
The English language quality is mostly fine; however, there are some issues with unclear use of non-synonymous words -- notably "homemaker" vs. "head of household", which mean different things in English.
Author Response
We would like to express our gratitude for your willingness to review this manuscript. Please find the detailed responses below in the attachment, along with and the corresponding revisions and corrections.

Round 2
Reviewer 1 Report
Comments and Suggestions for Authors
I would like to express appreciation to the authors for considering my comments thoroughly. However, there are still points that may benefit from further refinement.
-Regarding the conclusion, the assertion that "Implementing One Health strategies, including educational campaigns, income increase, and sustainable land-use practices, is crucial for effective control and prevention of CVL in urban areas," appears to extrapolate beyond the scope of the study's findings. The conclusion should be objective and aligned with the identified results.
-The points addressed in response to Comment 1 should be integrated into the main body of the article (into the methods, results, and discussion sections). Additionally, supplementary materials should be appropriately cited and explained within the text.
-Similar considerations should be applied to the refusals. The absence of detailed information and its implications should be acknowledged and discussed within the limitations section.
Author Response
Response to Reviewer 1 Comments |
||
1. Summary |
||
|
We would like to extend our sincere gratitude for your detailed and thoughtful review of our manuscript. Your comments and suggestions were extremely helpful and have greatly contributed to the improvement of our work. |
||
2. Questions for General Evaluation |
Reviewer’s Evaluation |
Response and Revisions |
|
Does the introduction provide sufficient background and include all relevant references? |
Yes |
We are grateful for the insights offered in the review. |
|
Are all the cited references relevant to the research? |
Yes |
We are grateful for the insights offered in the review. |
|
Is the research design appropriate? |
Can be improved |
Further to our previous communication, we have provided more detailed information concerning the aforementioned gaps in the design, specifically in relation to the samples of dogs discussed in the revision. |
|
Are the methods adequately described? |
Can be improved |
Further to our previous communication, we have provided more detailed information concerning the aforementioned gaps in the design, specifically in relation to the samples of dogs discussed in the revision. |
|
Are the results clearly presented? |
Can be improved |
In order to substantiate our findings, we have included additional information regarding the procedures performed and the associated limitations. |
|
Are the conclusions supported by the results? |
Can be improved |
In light of the findings presented in our manuscript, we have revised our conclusion to align with the results. |
3. Point-by-point response to Comments and Suggestions for Authors |
||
|
Comments 1: I would like to express appreciation to the authors for considering my comments thoroughly. However, there are still points that may benefit from further refinement.
-Regarding the conclusion, the assertion that "Implementing One Health strategies, including educational campaigns, income increase, and sustainable land-use practices, is crucial for effective control and prevention of CVL in urban areas," appears to extrapolate beyond the scope of the study's findings. The conclusion should be objective and aligned with the identified results. |
||
|
Response 1: Dear Reviewer, we are grateful for your comments, which help us to enhance the quality of the manuscript. We have aligned the highlighted sentence with the scope of our results. |
||
|
Comments 2: The points addressed in response to Comment 1 should be integrated into the main body of the article (into the methods, results, and discussion sections). Additionally, supplementary materials should be appropriately cited and explained within the text. |
||
|
Response 2: In accordance with the suggestions, the manuscript was modified to include the terms of the proposed topics. We have included the details: In Material and Methods, and in line 129: “The study included only dogs that were domiciled and for which the owner had consented to the sampling by the surveillance service. The georeferencing of the data (sampling of notification forms) was random. The number of forms was not the same for positives and negatives (proportion of sampled forms from the total and total number of forms sampled) because there are many more negative results than positive ones. The purpose of sampling the negative forms was to test whether areas without reactive dogs were non-reactive due to a lack of sampling or a true absence. Consequently, approximately 20% of the total sample was included, which involved a considerable effort of georeferencing approximately 2000 addresses.” In results in line 249: The sampling outcome is presented in the map in Figure S1, which depicts the density of reactive and non-reactive dogs. It can be observed that the density of negative dogs is largely consistent the same outside of foci, which suggests a random sampling. Additionally, it can be observed that the higher density of negative dogs is similar to the density of positive dogs, which, according to the LIRAa protocol are the foci, the most densely sampled areas. And in discussion in line 443: “Within the foci, the sampling is more thorough, until completely negative streets are found. This leads to the following consequence: once the relative risk (positives/negative) is considered, and that these higher risk zones are located within the foci, which are thoroughly sampled using the same protocol, our risk estimate is likely unbiased.” |
||
|
Comments 3: Similar considerations should be applied to the refusals. The absence of detailed information and its implications should be acknowledged and discussed within the limitations section. |
||
|
Response 3: Dear reviewer, we agreed with the comment in question. In line 447, we have included the following sentence: “Furthermore, only those households that consented to the sampling process conducted by the ESMZ were included in the study. Refusals were not provided to us, but it is possible that this data may alter the proportion of relative risk without altering the main results and conclusion.” |
||

Reviewer 2 Report
Comments and Suggestions for Authors
Thank you for addressing my comment, the manuscript is acceptable in this form with minor changes to the English language (typos).
Comments on the Quality of English LanguageSee above, some minor typos that can be corrected in production, but nothing serious.
Author Response
Thank you very much for your thorough review of our manuscript and for the insightful suggestions and comments. We greatly appreciate the time and effort you dedicated to improving the quality of our work. Your feedback has been invaluable in enhancing the clarity and robustness of our study.
